# Possible Association of Energy Availability with Transferrin Saturation and Serum Iron during Summer Camp in Male Collegiate Rugby Players

**DOI:** 10.3390/nu13092963

**Published:** 2021-08-26

**Authors:** Madoka Tokuyama, Jun Seino, Keishoku Sakuraba, Yoshio Suzuki

**Affiliations:** 1MORINAGA & Co., Ltd., 5-33-1 Shiba, Minato-ku, Tokyo 108-8403, Japan; maaaaado_11@icloud.com (M.T.); seino.jun.ka@u.tsukuba.ac.jp (J.S.); 2Graduate School of Health and Sports Science, Juntendo University, Inzai 270-1695, Japan; sakuraba@kf6.so-net.ne.jp; 3R&D Center for Smart Wellness City Policies, Faculty of Health and Sport Sciences, University of Tsukuba, Otsuka, Tokyo 112-0012, Japan

**Keywords:** body weight, exercise, football, hepcidin, weight loss

## Abstract

Low energy availability (LEA) may persist in rugby players. However, timely assessment of energy balance is important but is difficult. Therefore, a practical index that reflects energy availability (EA) is essential. A total of 19 male college rugby players participated in a 2-week pre-season summer camp. Their blood sample was collected after overnight fast prior to (Pre), in the middle (Middle), and after (Post) the camp. Their physical activity in the first half of the camp was calculated using the additive factor method in the forwards (FW; numbers 1–8) and backs (BK; numbers 9–15). The participants were categorized as tight five (T5; numbers 1–5), back row (BR; numbers 6–8), and BK for analysis. All the participants lost weight during the camp (range: from −5.9% to −0.1%). Energy balance in the first half of the camp was negative. Transferrin saturation (TSAT) and serum iron levels significantly decreased to half, or even less, compared with the Pre levels at week 1 and remained low. The changes in TSAT and serum iron levels exhibited a significant positive correlation with the changes in body weight (R = 0.720; R = 0.627) and with energy intake (R = 0.410; R = 461) in T5. LEA occurs in rugby summer camp but is difficult to assess using weight change. Alternately, TSAT and serum iron levels after overnight fast may be better predictors of LEA.

## 1. Introduction

Rugby is a contact sport that involves intermittent high-intensity anaerobic exercise followed by prolonged low-intensity aerobic exercise [1,2]. The average running distance per game is 8503 ± 631 m [3], and the energy expenditure is reported to be 24 kcal/min [4]. Therefore, it is a sport that results in high energy expenditure.

Rugby is also characterized by different competitive characteristics depending on the position. Forwards (FW; numbers 1–8) are classified into front rows (FR; numbers 1–3), second rows (SR; numbers 4–5), and back rows (BR; numbers 6–8) [5,6,7,8]. FR and SR are also combined as tight five (T5; numbers 1–5), as they are all involved in the activity of gaining or retaining possession of the ball and project similar horizontal forces, especially during scrums [9,10,11,12]. In addition, T5 players are involved in high-intensity and heavy-contact activity during the scrum [13]. In contrast, BR players have fewer set plays, such as rucks and mauls, than T5, and typically need to sprint and tackle in dense attacks [6,7,14,15]. For BR players, the speed of force rise and sprinting ability are more critical than those for T5. However, BR players are not required to be as heavy or strong as the T5 players [9]. BK players would cover more distance than FW players and require more running speed and acceleration ability [16]. Therefore, team practices and training sessions are carried out by player position [9], and the energy expenditure varies by player position, depending on the frequency of sprints, tackles, steps, and scrums.

The energy expenditure of rugby players is also affected by the game season (pre-season, in-season, off-season) and the physique of the player [17,18]. In addition, there are variations in the energy requirements due to age, gender, growth, and maturity; thus, the total energy expenditure of rugby players widely varies among individuals [19]. Due to the varied athletic characteristics required and the difficulty in determining the appropriate dietary intake to match energy expenditure, it is understood that rugby players have low energy availability (LEA).

Prolonged LEA can reduce metabolic efficiency and increase the risk of fatigue fractures, as well as adversely affect protein synthesis, cardiovascular and psychological health, and physical growth and development [20]. This problem has been shown to be a threat to male and female athletes, especially for endurance and weight class athletes [21,22].

One of the factors that can lead to LEA is the difficulty in assessing the state of EA, including the assessment of energy intake and expenditure. A review in 2018 noted the lack of reports that measure the dietary intakes of elite rugby players in general [23]. A recent study of young professional rugby players in pre-season [24] is believed to be the first report that has investigated energy balance (intake and consumption simultaneously). The report raised the issue of a large negative energy balance, wherein the total energy intake was 3996 kcal/d, whereas the total energy expenditure was 4385 kcal/d. This report showed the importance and at the same time difficulty of achieving a regular and rapid assessment of energy balance. Although dietary assessment tools and wearable technology have been validated for rapid and appropriate assessment of energy balance in response to changes, the determination of individual energy intake in athletes is challenging [24]. As aforementioned, a quick, accurate, and efficient method for assessing the EA status, including energy balance, is required.

This study proposes the use of transferrin saturation (TSAT) and serum iron as indicators of EA in the short term (1–2 weeks). In this study, a registered dietitian accompanied elite university rugby players during their entire pre-season training camp to assess their dietary intake. Their hematological and biochemical parameters were analyzed. In addition, coaches shared with us in real time the team schedule and the extent of practice sessions. Weight loss was observed in all the subjects before and after the training camp period. This indicates that they may have fallen into LEA. Therefore, this study aimed to find a rapid and practical indicator of EA status from the parameters obtained.

## 2. Materials and Methods

### 2.1. Participants and Study Design

A total of 30 male collegiate rugby players participated in a summer camp held from 9 August to 24 August 2017. The participants arrived at their own timing by 8 August after completing their university’s first semester exams.

The first half (9 August to 14 August) of the training session mainly consisted of group training divided into forward and back units, whereas the latter half (16 August to 22 August) mainly consisted of open games with other university teams. There was a day off on 15 August.

This study investigated the players’ conditions from 9 August to 22 August. Their diet was surveyed by a registered dietitian. Blood tests were conducted on 9 August (Pre-training camp, hereafter referred to as Pre), 15 (Middle of the training camp, hereafter referred to as Mid), and 22 (Post-training camp, hereafter referred to as Post). The players’ body weights were measured Pre and Post the study period.

Until 15 August (Middle), six players (five due to injury and one due to fever) dropped out of the training program. Another five players (two due to injury, one due to fever and diarrhea, and two due to elevation to the national U-20 team) dropped out by 22 August (Post). The 11 players who had dropped out were excluded from the analysis. The position and characteristics of the remaining 19 participants who completed the training program are presented in Table 1. In this study, the position was categorized as T5 (numbers 1–5; *n* = 7), BR (numbers 6–8; *n* = 4), and BK (numbers 9–15; *n* = 8), as the training program was designed separately for FW (T5 and BR) and BK; from previous studies [9,10,11,12], we determined that the characteristics of T5 and BR are different.

This study was originally planned and conducted as a controlled trial to assess the effect of bovine lactoferrin supplementation. However, the intervention (lactoferrin or placebo) did not make any difference in body weight, hematological and biochemical parameters, and dietary intake (data not shown); therefore, the data were analyzed together without distinction by the intervention.

The study protocol was approved by the Ethics Committee of Juntendo University Graduate School of Health and Sports Science (Approval number: 29–62) and was conducted in accordance with the ethical standards of the 1964 Declaration of Helsinki and its later amendments or comparable ethical standards. The study protocol was registered to UMIN Clinical Trials Registry (UMIN-ID: UMIN000028554).

### 2.2. Dietary Instructions

To prevent LEA and protein insufficiency, the players were instructed to take at least 50 kcal/kg BW/day of energy [25] and 2 g/kg BW/day of protein [26] with carbohydrate in the range of 8–12 g/kg BW/day [26]. Iron intake of 8 mg/day was considered adequate [26].

### 2.3. Dietary Survey

The dietary survey was conducted from 9 August to 21 August 2017, by dietary record.

Meals were served in the facility’s cafeteria three times a day (breakfast, lunch, and dinner). The main dishes were common for each player. Cooked rice and side dishes were free for each participant to consume. Each participant weighed the amount of cooked rice and took a picture of it with the weight indicated on the scale. Side dishes were photographed prior to consumption. The photos were submitted to the dietitian for each meal. This enabled us to determine the intake of all main courses, cooked rice, and side dishes. The weights of the main dishes were measured separately. Moreover, the weights of the side dishes were calculated if they were served in portions, and otherwise estimated from photographs. The dietitian instructed the participants to consume all the food provided, and the dietitian and team managers confirmed that there was no leftover food.

In addition to meals, the athletes were provided with 2 L of milk and 3 bananas daily. Protein supplement was also provided for each participant. The food supplements provided to all athletes was consumed without any leftovers. All the foods provided were reported to the dietitian along with the type and weight.

There were two drink vending machines in the facility, and bottled or canned beverages could be freely purchased. Purchased beverages were photographed and submitted to the dietitian.

After practice, the participants were free to shop at a convenience store outside the facility. Each time, any food purchased was photographed and submitted to the dietitian prior to consumption.

The participants had no other means of obtaining food other than the aforementioned. However, players were free to consume water, including during practice, and the amounts were not reported.

As described above, the type and amount of food, other than water, consumed by each participant was ascertained without omission.

The energy and nutrient intake of each athlete was calculated using the nutrition calculation software Healthy Maker Pro 501 (Mushroomsoft Co., Ltd., Okayama, Japan). Daily intakes were expressed per body weight (kg) for energy, protein, fat, and carbohydrates and then mg for iron.

However, we were not able to determine the amount of food consumed on 14 and 21 August as dinners were served buffet style, and on 15August, which was a day off, the participants visited downtown. Hence, these 3 days were excluded from the days of assessment.

### 2.4. Blood Analysis

Blood was collected from the cubital vein in the morning before eating breakfast.

Serum albumin, creatine kinase (CK), ferritin, iron, and total iron-binding capacity (TIBC) were analyzed using a JCA-BM8060 automated analyzer (Japan Electron Optics Laboratory Ltd., Tokyo, Japan). Hematologic parameters, including white blood cell, red blood cell, hemoglobin (Hb), hematocrit (Hct), mean corpuscular volume, mean corpuscular hemoglobin (MCH), MCH concentration, and platelet (PLT), were analyzed using a Sysmex XE 2100 automated hematology analyzer (Sysmex Corporation, Kobe, Japan). A commercial laboratory service, Hoken Kagaku Inc. (Yokohama, Japan), conducted the analyses and provided reference ranges of healthy Japanese male adults.

Serum hepcidin was analyzed using a commercially available enzyme immunoassay kit (Quantikine ELISA Human Hepcidin Immunoassay, R&D Systems, Minneapolis, MN, USA). The sample range was provided by the manufacturer.

TSAT was calculated as serum iron/TIBC. The reference range was obtained from the MDS Manual Professional Edition.

### 2.5. Physical Activity

The first half of the training camp (9 August to 14 August) consisted mainly of group training, whereas the second half (16 August to 22 August) consisted of open games with other university teams.

The practice program for the first half was designed separately for the FW and BK players, and the duration of each activity was visually monitored by the researcher who recorded the type of practice or training and the time it took place. From these records, the average amount of energy consumed by the FW and BK players was estimated using the additive factor method [27,28,29].

The type of practice was selected as the equivalent from the “Revised METs (METs) Table of Physical Activity” (Ministry of Health, Labour and Welfare of Japan, 2011), which was based on Ainsworth’s reports [28,29]. The amount of each physical activity (METs × h) was calculated by multiplying the METs value of each practice by the time conducted. The amount of activity other than practice was calculated from the daily schedule (recreation, meal, and sleep) in the same manner.

The energy expenditure due to exercise (kcal/kg BW) and the energy expenditure due to activities other than exercise (kcal/kg BW) were calculated by multiplying the sum of each physical activity by 1.05 (METs × h × 1.05).

The second half of the training camp was mostly open games; thus, we were unable to record and track the amount of activity for each participant.

### 2.6. Statistical Analysis

Data are expressed as median with range or estimated marginal mean with standard error. The changes in body weight were compared using Wilcoxon signed-rank test. The changes in hematological and biochemical parameters were analyzed using the generalized estimating equation with *p*-value adjusted via the Bonferroni correction. Dietary intakes between the positions were also compared using the generalized linear model with *p*-value adjusted via the Bonferroni correction.

The changes in body weight, TSAT, and serum iron were calculated as (Post − Pre)/Pre (%), and the correlation with the intake of energy, protein, and iron was assessed using Spearman’s correlation coefficient (r_s_).

SPSS ver. 19 (Japan IBM, Tokyo, Japan) was used for the analysis. Statistical significance was set to *p* < 0.05, and for Pearson’s correlation coefficient, it was set to |r| > 0.3 [30].

## 3. Results

### 3.1. Body Weight

All the participants lost weight during the study period. The weight loss was significant for T5 (*p* < 0.05) and BK (*p* < 0.05), and the mean decrease (−1.7% and −1.9%) was similar, whereas the mean change in BR was smaller (−1.0%). The change in the BR mean was not significant due to the small sample size (*n* = 4) (Table 1).

### 3.2. Biochemical and Hematological Parameters

For players at all positions, TSAT (reference range: 20–50%) and serum iron (reference range: 45–200 µg/dL) levels significantly decreased at Mid to half or less compared with the Pre levels and then remained low until Post. The mean TSAT levels were within the reference range at Pre but remained near or below the lower limit of the reference range at Mid (Figure 1).

In T5 and BK, TIBC was significantly higher at Mid than Pre and lower at Post, but the changes were within the reference range (245–385 µg/dL). The means of ferritin were within the reference range (45–200 µg/dL), while they raised at Mid in BR and BK (Figure 1).

The mean CK was above the reference range (50–250 U/L) for players at all positions throughout the camp. The other parameters were all within the reference range (Appendix A).

### 3.3. Dietary Intake and BMI for Each Position

During the camp (from Pre to Post), 15 (out of 19) participants, except for 4 in T5, consumed more than the goal for energy (50 kcal/kg/d), but none of them achieved that for protein (2 g/kg/d). Meanwhile, 14 of the 19 participants consumed more than 8 g/kg/d of carbohydrates. All of them consumed more than 8 mg/d of iron.

Dietary intake differed according to the positions of the players (Table 2). The energy, protein, and fat intakes of T5 for the entire training camp period (Pre to Post) were significantly lower than BR and BK. The mean carbohydrate intake was also the smallest for T5 and significantly different from BK. There were no significant differences in the iron intake between the positions.

The intake differences among positions were more apparent in the second half of the camp (Mid to Post) than the first half of the camp (Pre to Mid). BK and BR consumed significantly more energy, protein, fat and carbohydrate than T5. In the first half of the camp (Pre to Mid), BK showed greater consumptions in energy, protein, fat and carbohydrate than T5. In contract, there were no differences in iron intake between positions throughout the camp.

### 3.4. Physical Activity

The average physical activities were calculated for FW and BK for the first half of the summer camp, since the practice program was designed separately for FW and BK. The exercises were categorized into yoga, team practice, stretch, unit training, skill training, core exercise, weight training, strength and fitness, fitness, cap run, pool, game, and personal training (Appendix A).

The average daily physical activity was 64.0 METs × h/kg (67.2 kcal/kg) and 71.0 METs × h/kg (74.6 kcal/kg) for FW and BK, respectively. Exercise accounted for 63% for FW and 68% for BK.

In the second half of the summer camp, we could not calculate the physical activity as the program mainly comprised of games, and the activity of each participant was not recorded.

### 3.5. Energy Balance

In the first half of the camp, the energy balance was negative, i.e., energy expenditure was greater than energy intake, for all positions (Figure 2). The EAs (“energy intake”—“energy expenditure from exercise”) for T5, BR, and BK were 10.4, 15.8, and 9.1 kcal/kg/d, respectively. The EA status was smaller than the proposed lower limit of 30 kcal/kg FFM/d [26], even considering the difference between BW and FFM.

### 3.6. Correlation of BW Change with Blood Parameters and Intakes

The changes in body weight during the camp demonstrated positive correlations with the changes in TSAT and serum iron levels. Looking at the values for the corresponding positions, the correlation was significant for T5 (r_s_ = 607) and BK (r_s_ = 714) for TSAT, and T5 (r_s_ = 607) and BK (r_s_ = 619) for serum iron (Figure 3A,B).

No significant correlation was observed between weight change and intakes of energy, protein, and iron. However, looking at the corresponding positions, positive correlations were observed for protein (r_s_ = 0.393) and iron (r_s_ = 464) for T5, whereas negative correlations were observed for BR and BK (Table 3).

### 3.7. Correlation of TSAT and Serum Iron with Intakes

The changes in TSAT and serum iron levels were not positively correlated with energy, protein, or iron intake. However, by position, T5’ protein intake exhibited a significant positive correlation with TSAT (r_s_ = 0.536) and with serum iron (r_s_ = 0.536). However, negative correlations were observed for BR and BK (Table 3).

## 4. Discussion

During the intensive training period for rugby, EA needs to be satisfied to maximize adaptation to training [26]. It has been reported that the daily energy intakes of Australian professional rugby players during the pre-season were 43.8 and 48.4 kcal/kg for FW and BK, respectively [31]. It has also been reported that the pre-season daily energy intakes of rugby players who participated in the All-Japan University Championship were 41.0 and 40.8 kcal/kg for FW and BK, respectively [32]. In this study, the daily energy intakes in the first half of the training camp were 53.0, 58.4, and 59.9 kcal/kg for T5, BR, and BK, respectively, which were greater than those reported in previous studies. However, as energy expenditure exceeded the intake, the energy balance became negative. In addition, all participants lost weight during the camp. However, the relationship between daily energy intake and weight change was not clear. When assessed by position, a significant positive correlation was observed in T5, possibly due to the smallest intake with a large variation, whereas significant negative correlations were observed for other positions. In addition, the change in body weight was small. Therefore, the change in body weight was not a good indicator for the assessment of EA during the training camp.

Furthermore, dehydration reduced body weight by 1.5% during high-intensity rugby training, with a maximum loss in body weight of up to 2.4% in some players [33]. Weight loss due to dehydration has been reported to occur prior to, during, and several days post-games [34]. The weight change has been proposed as an indicator of dehydration [35]. Therefore, as dehydration also affects, it is difficult to assess EA status during the rugby camp using weight change as an indicator.

Furthermore, negative energy balance can also lead to metabolic adaptation and may not necessarily reflect immediately as a change in body weight [26]. Therefore, a sensitive index that can be used as an early indicator of EA status is essential. In this study, we examined the indices associated with changes in body weight. The changes in body weight correlated positively with the changes in TSAT and serum iron levels. Moreover, the maximum change in TSAT and serum iron levels was about −80%, which was much higher than the change in body weight (mostly less than −3%). Additionally, the change was apparent in a week. Therefore, the changes in TSAT and serum iron levels can be used as sensitive indices to predict changes in body weight.

In addition, changes in TSAT and serum iron levels demonstrated significant positive correlations with protein intake but not with iron intake in T5, whose energy intake was small but highly variable. Since protein is more used as a fuel source in LEA, the correlation was possibly more apparent than with energy intake. Therefore, changes in TSAT and serum iron levels can be used as indicators for EA status.

Approximately 1% of red blood cells (RBCs) turnover daily, this turnover requires energy for the novel RBC production as well as for the degradation of old RBCs. Meanwhile, this turnover is a metabolic process that is thought to be suppressed when energy intake is low. Therefore, LEA could affect the RBC turnover. However, we could not find any literature showing that short-term LEA affects RBC turnover. Therefore, we could not discuss the theoretical mechanism by which TSAT and serum iron reflect EA status. However, the relationship between hepcidin and EA status has been discussed recently.

Hepcidin, whose secretion from the liver is induced by exercise, suppresses ferroportin and inhibits the release of intracellular iron to circulation [36]. However, in overnight fast serum, hepcidin was significantly elevated in female long-distance runners during the competitive season compared with the baseline training, wherein the serum iron levels and TSAT were also elevated [37]. Female long-distance runners were also reported to have lower serum iron level after 8 weeks of training, with reduced levels of hepcidin observed in runners with less than 20 ng/mL of ferritin [38]. Exercise has not been observed to result in elevated hepcidin and decreased serum iron level after overnight fast. It has been reported that hepcidin elevated for 3 h after a single graded exercise test on treadmill and subsequently returned to baseline level 24 h post-exercise, whereas the mean serum iron levels 24 h post-exercise were lower than the baseline values [39]. This indicates that serum iron can reflect the cumulative effect of exercise-induced hepcidin within a day. Furthermore, hepcidin has been shown to be increased after 4 days of military training, with the concentration correlating negatively (r = −0.43) with the energy balance [40]. Recently, hepcidin was proposed as an indicator of LEA [41]. In the present study, hepcidin did not show any significant changes and remained within the reference range during the camp. Moreover, no significant correlation was observed with serum iron. Serum iron and TSAT in overnight fast serum may be more suitable than hepcidin to assess EA status, if they reflect the cumulative effect of exercise-induced hepcidin.

This study has a few limitations. First, dietary survey was conducted for each participant, but the activity level was not individually surveyed. Therefore, we were able to estimate the amount of average physical activity in the first half of the camp, when the program was organized by position, but could not assess the amount of activity of individual participant, especially in the second half of the camp. However ideally, energy expenditure should have been assessed using the doubly labeled water method. Second, body weights were not measured daily; thus, we were unable to examine the relationship between EA and weight in the first half of the camp. Furthermore, the lack of body composition measurements made it impossible to determine changes in fat mass and lean mass. In this study, we were able to demonstrate the relationship between LEA, TSAT, and serum iron in a rugby camp with 19 participants, but the number of participants at each position was limited, especially BR with only 4 participants. In addition, this study was originally designed as an intervention study. Therefore, nine players ingested bovine lactoferrin during the camp. Although significant difference was not observed in any of the parameters reported in this study, the possibility was not deniable that existence of responder/non-responder to bovine lactoferrin affected the results. Therefore, a future study with a larger number of participants will be necessary to generalize the findings of this study.

## 5. Conclusions

LEA occurs in rugby pre-season camps. However, it is difficult to estimate the energy requirements of rugby players with large individual differences. In this study, a registered dietitian accompanied the rugby players during the pre-season camp to ascertain their intakes and encourage dietary intake; however, as a result, LEA occurred, which could not be assessed during the camp, and therefore, the condition could not be improved. On the other hand, we found that the TSAT and serum iron levels changed dramatically within a week and may be indicative of the EA status. Blood sampling post-overnight fast is a practical procedure even in rugby camps. TSAT and serum iron levels can serve as effective indicators of EA status and may be useful for managing, controlling, and improving the EA status in rugby.

## Figures and Tables

**Figure 1 nutrients-13-02963-f001:**
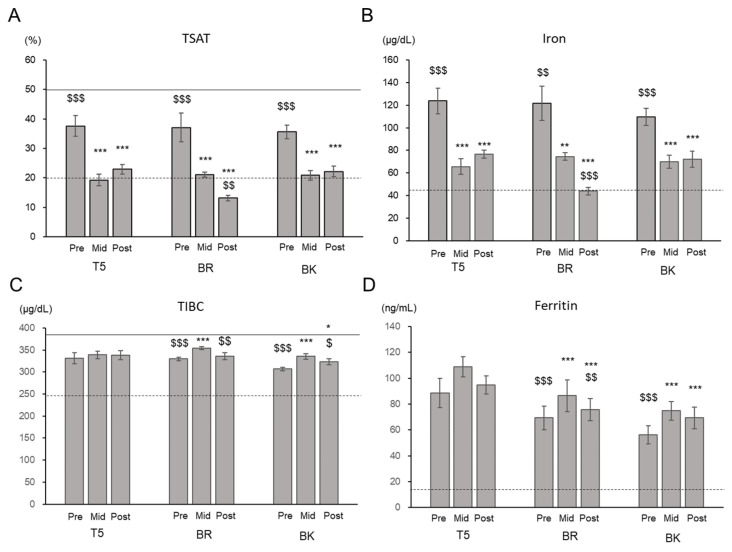
Changes in iron parameters. (**A**) Transferrin saturation (TSAT), (**B**) serum iron level, (**C**) total iron-binding capacity (TIBC), and (**D**) ferritin level. T5, Tight-5 (numbers 1–5); BR, Back row (numbers 6–8); and BK, Backs (numbers 9–15). Bars and error bars represent estimated marginal means and standard error, respectively. The solid and dashed lines indicate the upper and lower limits of the normal range, respectively. * *p* < 0.05, ** *p* < 0.01, *** *p* < 0.001 vs. Pre; $ *p* < 0.05, $$ *p* < 0.01, $$$ *p* < 0.001 vs. Mid.

**Figure 2 nutrients-13-02963-f002:**
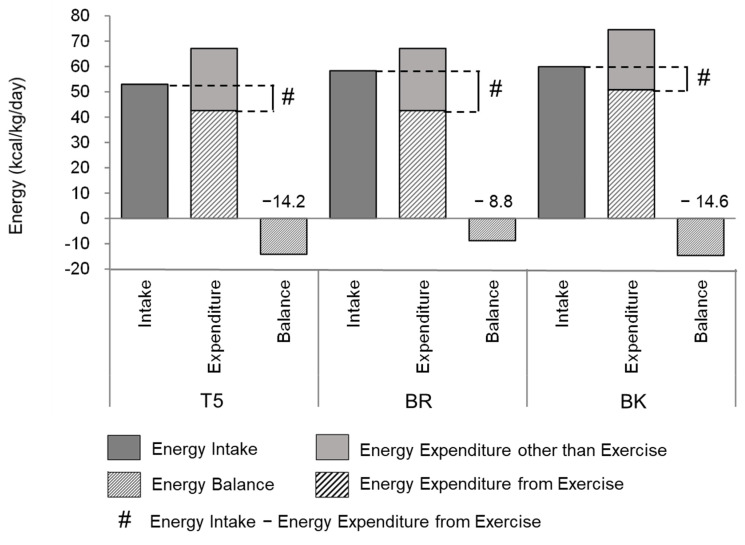
Energy balance during the first half of the training camp. T5, Tight-5 (numbers 1–5); BR, Back row (numbers 6–8); and BK, Backs (numbers 9–15). Energy availabilities (#) for T5, BR, and BK were 10.4, 15.8, and 9.1 kcal/kg/d, respectively.

**Figure 3 nutrients-13-02963-f003:**
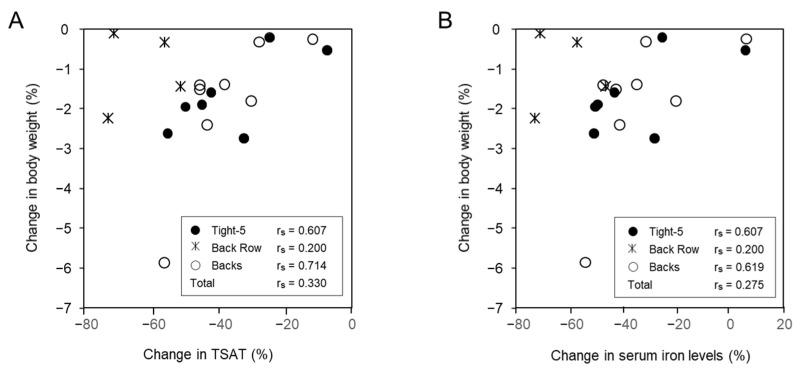
Correlations of changes in body weight with changes in (**A**) transferrin saturation (TSAT) and (**B**) serum iron level. Change was calculated as (Post − Pre)/Pre (%). Tight-5 (numbers 1–5); Back row (numbers 6–8); and Backs (numbers 9–15).

**Table 1 nutrients-13-02963-t001:** Characteristics of the participants.

		Age				Height				Body Weight									Change in Body Weight			
	*n*	Pre (year)				Pre (cm)				Pre (kg)				Post (kg)					(Post − Pre)/Pre (%)			
		Median	Range			Median	Range			Median	Range			Median	Range			*p*	Median	Range		
T5	7	21	20	-	22	181	165	-	188	94.9	92.9	-	112.6	93.1	92.0	-	110.8	0.004 **	−1.9	−2.7	-	−0.2
BR	4	21	21	-	22	179	176	-	183	89.4	85.5	-	98.2	88.6	85.4	-	96.0	0.138	−0.9	−2.2	-	−0.1
BK	8	21	20	-	22	174	170	-	180	86.0	79.0	-	92.7	84.2	77.1	-	92.4	0.021 *	−1.5	−5.9	-	−0.3

T5, Tight-5 (numbers 1–5); BR, Back row (numbers 6–8); and BK, Backs (numbers 9–15). * *p* < 0.05, ** *p* < 0.01.

**Table 2 nutrients-13-02963-t002:** Energy, energy source, and iron intake during the study period.

Energy and Nutrient	Goal	Position	*n*	Pre–Middle			Middle–Post			Pre–Post		
				EMM	SE		EMM	SE		EMM	SE	
Energy (kcal/kg/day)	>60	T5	7	53.0	1.7		45.8	1.5		49.4	1.5	
		BR	4	58.4	2.3		55.1	2.0	***	56.7	2.0	**
		BK	8	59.9	1.6	**	56.0	1.4	***	58.0	1.4	***
Protein (g/kg/day)	>2	T5	7	1.8	0.1		1.5	<0.1		1.6	<0.1	
		BR	4	1.8	<0.1		1.8	<0.1	***	1.9	<0.1	***
		BK	8	2.0	0.1	*	1.8	<0.1	***	1.9	<0.1	***
Fat (g/kg/day)		T5	7	1.1	<0.1		1.0	<0.1		1.1	<0.1	
		BR	4	1.2	<0.1		1.2	<0.1	**	1.2	<0.1	***
		BK	8	1.3	<0.1	***	1.2	<0.1	***	1.2	<0.1	***
Carbohydrate (g/kg/day)	8–12	T5	7	8.5	0.4		7.5	0.3		8.0	0.3	
		BR	4	9.5	0.5		8.9	0.4	**	9.2	0.4	
		BK	8	9.7	0.3	*	9.2	0.3	***	9.5	0.3	**
Iron (mg/day)		T5	7	13.3	0.6		11.1	0.6		8.0	0.3	
		BR	4	12.9	0.7		12.5	0.7		9.2	0.4	
		BK	8	12.8	0.5		12.2	0.5		9.5	0.3	

EMM, estimated marginal mean; SE, standard error; vs. T5: * *p* < 0.05, ** *p* < 0.01, *** *p* < 0.001; T5, Tight-5 (numbers 1–5); BR, Back row (numbers 6–8); and BK, Backs (numbers 9–15).

**Table 3 nutrients-13-02963-t003:** Spearman’s correlation coefficient between the change in body weight, TSAT, and iron versus change in intakes of energy, protein, and iron.

Position	*n*	Change ^$^	Energy (kcal/kg)		Protein (g/kg)		Iron (mg/day)	
Total	19	Body Weight	−0.065		0.095		−0.144	
		TSAT	−0.235		−0.347		−0.288	
		Serum Iron	−0.070		−0.168		−0.339	#
T5	7	Body Weight	0.107		0.393	*	0.464	*
		TSAT	0.179		0.536	**	−0.071	
		Serum Iron	0.179		0.536	**	−0.071	
BR	4	Body Weight	−0.200		−0.800	###	−0.200	
		TSAT	−1.000	###	−0.400	#	−1.000	###
		Serum Iron	−1.000	###	−0.400	#	−1.000	###
BK	8	Body Weight	−0.667	##	−0.524	##	−0.286	
		TSAT	−0.333	#	−0.595	##	−0.571	##
		Serum Iron	−0.214		−0.476	#	−0.619	##

^$^ Change represents (Post − Pre)/Pre; * r_s_ > 0.3, ** r_s_ > 0.5; # r_s_ < 0.3, ##, r_s_ < −0.5, ### r_s_ < −0.7; T5, Tight-5 (numbers 1–5); BR, Back row (numbers 6–8); and BK, Backs (numbers 9–15).

## Data Availability

The data presented in this study are available on request from the corresponding author. The data are not publicly available due to ethical restrictions.

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
