# Peer review of "Possible Association of Energy Availability with Transferrin Saturation and Serum Iron during Summer Camp in Male Collegiate Rugby Players"

_nutrients, 2021, doi:10.3390/nu13092963_

Round 1
Reviewer 1 Report
I read with interest your manuscript . It is well-written, interesting general information based on recent literature, good design, strong statistics, detailed results, important conclusion and recent and valuable references (proper cited). I recommend to move the last sentences before Conclusions, into Conclusion section (to finalize it with recommendations based on your findings). I found it interesting for readers, based on a good presentation and correct English language, a footprint of novelty, and recent references cited as well.
Author Response
Dear Reviewer-1,
We appreciate your kind and valuable comments on our manuscript.
According to your suggestion, we moved the last paragraph of the Discussion section to Conclusion section.
Sincerely,
Yoshio Suzuki, PhD
Reviewer 2 Report
The manuscript is very well written. Below are my comments
Introduction
- I appreciate the thoroughness of the authors at describing why LEA is an issue in rugby players and how energy needs are different by position.
- The authors need to provide some sort of justification as to why TSAT was used to measure LEA. What are the theoretical underpinnings behind the use of this method.
Methodology:
- The authors were extremely detailed in the description of their procedures. Their thoroughness is appreciated. I also appreciate the candor regarding the fact that this was initially a placebo controlled trial.
- I understand that there weren't any significant differences between interventions. With such small n's did the authors try examining the data at an individual level? Our lab is showing that inter-individual differences in interventional responses can sometimes average out to not having significant findings (we have seen this in nutrition and exercise related studies). This can lead to errors in reporting results as the hyper responders can average out the non or negative responders. This may be an issue to examine and if there are individual differences by intervention then you need to report that as one of your limitations.
- Was the data normally distributed? With such small n's it may be challenging to justify using parametric analyses, especially in a repeated measures case. I'd suggest the authors re-examine the normality of distribution and then use non-parametric analyses where required.
- You may also benefit from using a Spearman rho for your correlations as a n of 11 may not meet assumptions for normality.
Results
- I question the validity of many of these results as the used parametric analyses where non-parametric were needed. Using a Wilcoxon rank sum test you may have been to determine whether there was a significant difference between pre and post measures
Author Response
Our response is attached as a PDF.

Reviewer 3 Report
Optimizing dietary recommendations for athletes, including energy intake, are key to achieving good performance and satisfactory sports results. The search for effective ways to control the nutritional status and nutrition of athletes is an important element in creating training patterns and has practical justification in scientific research.
The manuscript did not discuss the possibility of assessing energy expenditure using, among others, indirect respirometry or devices measuring physical activity. In assessing the nutritional status, and thus also assessing the availability of energy from the diet, it is also important to assess body composition.
MATERIAL AN METHODS
- The authors state in the text of the manuscript that 19 rugby players participated in the study, but 15 athletes are listed in the group structure: FW; numbers 1–8 and BK; numbers 9–15.
- The values of the changes in body weight that were observed during the camp, given in Table 1, should be included in the "results" chapter.
- It would be worth providing the literature on the basis of which energy expenditure was calculated.
- Could it be possible to mark the statistical differences between the groups of players in figure 1 and describe them? Similarly, were there statistical differences in the intake of energy, protein, fat, carbohydrates and iron in the different groups of players after one week and two weeks of training camp? This information should be marked in table 2. Similarly, it should be stated, for example in figure 2, whether the energy balance and the value of available energy differ statistically between the groups of players.
- In addition to lactoferrin supplementation, have rugby players used any other dietary supplements?
DISCUSSION
- For a better discussion of the results, it may be worth comparing the degree of implementation of dietary recommendations by players from individual groups.
- The greater than expected energy expenditure of rugby players and a negative energy balance indicate that the assumed dietary recommendations did not correspond to the actual intensity of exercise. Does such a statement indicate the need to optimize dietary care to improve athletes' performance? In such a situation, do we need new indicators not directly related to the diet? Wouldn't it be enough to assess the energy expenditure with the available methods and adjust the dietary quality to them and control changes in the weight and body composition (including body water content) of players?
- Is there any other explanation for such significant changes in TSAT values as a result of intense exercise planned in the first week of the training camp? It is worth noting that its changes were not so visible in the second week of the training camp, which could be related to a different nature of physical activity, and not a change in energy availability. Is it possible that this is a process accompanying training adaptation?
- Was the correlation between EA and changes in TSAT values and blood iron concentration assessed - which could support the conclusion: "Overnight fast TSAT and serum iron levels may reflect EA status within a week"?

Author Response
Our response is attached as a PDF.

Round 2
Reviewer 2 Report
Thank you for taking the time to address my questions.
Below are some more issues with the statistics
- Spearmans Rho is presented as rs
- When you're presenting data with such small N's it is best not to take means (as outliers will throw those off). Instead it is best to present median
- For figure 1 since you are using non-parametric statistics, SPSS should give you an output with the data and you should be able to use those figures as small n's (especially with BK only having 4), it is challenging to present the readers with means and SD as they're not truly representative of the results. Please modify your figures and in your tables also present median values.
Author Response
Dear Reviewer-2,
I appreciate your prompt checking the resubmitted manuscript.
I agree that the mean and SD do not represent the data with small number.
As I used generalized linear model (including generalized estimating equation) to evaluate the P value, estimated marginal mean and se were outputted.
Before revising the manuscript, I would like to ask the followings 2 questions.
- Is it OK to use the estimated marginal mean and se instead of median and range (or quartiles) in revising Figure 1 and Tables 2 and Supplementary Table S1?
- Is it OK to revise Table 1 (characteristics of the participants) with median and range, because it aims to show the characteristics, not aiming to compare groups?
The editorial office requires to resubmit within 3 days.
Therefore, I would like to hear you instruction as soon as possible.
Sincerely,
Yoshio Suzuki